# Is the Use of Bisphosphonates Putting Horses at Risk? An Osteoclast Perspective

**DOI:** 10.3390/ani12131722

**Published:** 2022-07-03

**Authors:** Fernando B. Vergara-Hernandez, Brian D. Nielsen, Aimee C. Colbath

**Affiliations:** 1Department of Animal Science, Michigan State University, 474 S. Shaw Ln, East Lansing, MI 48824, USA; vergarah@msu.edu (F.B.V.-H.); bdn@msu.edu (B.D.N.); 2Department of Large Animal Clinical Sciences, College of Veterinary Medicine, Michigan State University, 736 Wilson Ave, East Lansing, MI 48864, USA

**Keywords:** bisphosphonates, osteoclast, bone, resorption, analgesic, anti-inflammatory, equine, juvenile, exercise

## Abstract

**Simple Summary:**

Bisphosphonates are a group of drugs that intervene in the bone resorption process, producing cellular death of osteoclasts. These drugs are used for skeletal conditions, such as osteoporosis in humans, and are available for veterinary medical use. Clodronate and tiludronate are bisphosphonates approved for the treatment of navicular syndrome in horses over four years old. However, these drugs are sometimes used in juvenile animals under exercise, where osteoclast activity is higher. Bisphosphonate use in juvenile and/or exercising animals could have adverse effects, including maladaptation to exercise or accumulation of microdamage. Furthermore, bisphosphonates can be bound to the skeleton for several years, resulting in a prolonged effect with no pharmaceutical reversal available. This review presents an overview of osteoclast function and a review of bisphosphonate characteristics, mechanisms of action, and side effects in order to contextualize the potential for adverse/side effects in young or exercising animals.

**Abstract:**

Osteoclasts are unique and vital bone cells involved in bone turnover. These cells are active throughout the individual’s life and play an intricate role in growth and remodeling. However, extra-label bisphosphonate use may impair osteoclast function, which could result in skeletal microdamage and impaired healing without commonly associated pain, affecting bone remodeling, fracture healing, and growth. These effects could be heightened when administered to growing and exercising animals. Bisphosphonates (BPs) are unevenly distributed in the skeleton; blood supply and bone turnover rate determine BPs uptake in bone. Currently, there is a critical gap in scientific knowledge surrounding the biological impacts of BP use in exercising animals under two years old. This may have significant welfare ramifications for growing and exercising equids. Therefore, future research should investigate the effects of these drugs on skeletally immature horses.

## 1. Introduction

Since their discovery in 1873, osteoclasts have been recognized for their bone resorption ability [1]. Osteoclasts’ resorption ability makes them a key cell for musculoskeletal development, bone metabolism, and bone repair throughout life [2]. Due to their intricate involvement in physiological processes, substantial pathologies are associated with overactive or impaired osteoclast function. In conditions where osteoclast resorption overcomes bone formation, it can be useful to decrease osteoclast activity through pharmaceutical interventions. Bisphosphonates (BPs), a class of drugs known to impair osteoclast function, have been available for human skeletal conditions for over 50 years and have been available in veterinary medicine for about 25 years [3].

In 2014, two bisphosphonates (Osphos^®^ and Tildren^®^) were approved by the United States Food and Drug Administration (FDA) to treat navicular syndrome in horses over four years of age [4]. However, it is unknown how these drugs affect juvenile animals under exercise, where skeletal adaptations depend upon normal bone metabolism and normal osteoclast function [5]. The objective of this review is to explore the available scientific literature regarding the origin of osteoclasts and their functions, how BPs affect these cells, the current use and side effects of BPs in humans and other animal models, and the potential negative effects of BPs use in the juvenile horses under exercise.

## 2. Origin of Osteoclasts

Osteoclasts’ precursors are found in the bone marrow and originate from the myeloid lineage [6]. Key molecules involved in osteoclastogenesis include members of the tumor necrosis factor family α (TNFα), such as the receptor activator of nuclear factor (NF)-ĸB (RANK), RANK ligand (RANKL), osteoprotegerin (OPG), and the macrophage-colony stimulating factor (M-CSF) [7]. In the presence of M-CSF and other growth factors, such as interleukin-3 (IL-3), osteoclast precursors proliferate and become preosteoclasts. These preosteoclasts fuse and generate mature multinucleated osteoclasts [8]. OPG serves as a negative feedback molecule for osteoclastogenesis, decreasing the RANKL function and osteoclast differentiation [6]. Osteoclastogenesis and subsequent osteoclast-mediated resorption are necessary for skeletal development, bone remodeling, and bone repair. Disruptions in osteoclastogenesis can lead to disease processes ranging from osteoporosis to osteopetrosis [8,9,10]. A summary of the main molecules involved in osteoclastogenesis are presented in Table 1.

## 3. Osteoclasts Are Critical for Bone Modeling and Remodeling

Osteoclast morphology plays an important role in their function; osteoclasts are multinucleated cells with an active “ruffled membrane” where resorption occurs [13]. When adjacent to the bone, the finger-like extensions of the cytoplasm adjacent to the bone produce a microenvironment through a proton pump, acidifying and demineralizing the bone matrix [14]. Alterations in osteoclast function can lead to critical bone diseases, including osteopetrosis [15], but may also have more subtle effects on bone modeling and remodeling.

Bone is a connective tissue with multiple roles, including mechanical support, protection, locomotion, mineral homeostasis, and endocrine functions [12,16,17]. Bones can be erroneously perceived as static tissue when, in fact, bones are constantly adapting to strain [14,18]. Bone adaptation is driven by two processes: bone modeling and remodeling [2]. Bone modeling is the process of mineral uptake and removal in growing organisms, leading to bone maturation. Meanwhile, bone remodeling is a process that takes place throughout the life of organisms, as the bone adapts to new mechanical loads and repairs microdamage, allowing the bone to reach a proper geometry [19]. In bone remodeling, bone resorption and bone formation are tightly coupled. Osteoclasts are recruited and activated, resulting in bone resorption and then undergoing apoptosis. Then, osteoblasts produce a new organic bone matrix, followed by mineralization [19].

## 4. Osteoclast-Modifying Drugs: Bisphosphonates

Bone modeling and remodeling can be affected by pharmacological interventions, including bisphosphonates. Bisphosphonates (BPs) vary in their chemical structure, complexity, and binding capacity to bone tissue [20]. BPs are chemically stable analogs of inorganic pyrophosphates (PPi). BPs are not subject to enzymatic hydrolysis, are resistant to high temperatures, are not biodegradable [21,22], and have a high affinity to bone hydroxyapatite (HAP) [20] (Figure 1).

Bisphosphonates have been classified into “generations” and mechanisms of action: first-generation BPs (i.e., clodronate), second-generation BPs (i.e., tiludronate and alendronate), and third-generation BPs (i.e., ibandronate and zoledronate). Newer generations have greater antiresorptive capabilities, which allows for a lower dose administration [22]. BPs are also classified according to their mechanism of action: simple or non-nitrogen-containing BPs (sBPs) and nitrogen-containing BPs (nBPs). Both BP types interfere with the osteoclast activity but with different relative potency [23]. The R_2_ side chain of BP structure determines the biological activity (Figure 1), and the presence of nitrogen atoms in the R_2_ side chain has shown a larger antiresorptive effect (Figure 2) [24]. Simple BPs (i.e., clodronate and etidronate) are metabolized in the cytoplasm and generate cytotoxic ATP analogs (5’-[β, γ-dichloromethylene] triphosphate), resulting in apoptosis due to a lack of free/functional ATP for cellular enzymatic function [20,23]. On the other hand, nBPs affect intracellular signaling through farnesyl diphosphate synthase (FPPS) inhibition [20,23]. This interferes with the prenylation of GTPase proteins, vital for functions such as the formation of the ruffled membrane, vesicular transportation, or apoptosis [24]. FPPS impairment also produces an accumulation of isoprenyl pyrophosphate (IPP), producing another ATP analog (1-adenosine-5′-yl ester 3-[3-methylbut-3-enyl] ester), causing a similar effect to sBPs [20,22,23,25]. Bone resorption, in turn, affects the bone formation phase due to the complex and coupled process of bone modeling and remodeling [26,27].

The pharmacokinetics (PK) of BPs depends on the route of administration. BPs are poorly absorbed orally, attributed likely to their low lipophilicity [28]. Oral bioavailability in humans has been estimated between 0.3% for pamidronate [29], 0.7% for alendronate [30], and 1–2% for clodronate [31]. Parenteral routes provide better and nearly complete absorption of BPs [28]. Initial tissue distribution depends on the protein-biding properties and will vary depending on blood pH, serum calcium, drug dosage, and species [28]. Though there is evidence of non-calcified tissue retention of BPs at high dosages [32,33], the probability of this occurring at therapeutic dosages is minimal [28]. BP distribution is not homogeneous in the skeleton and may be affected by sex and age. BPs tend to bind to the trabecular bone because of larger amounts of bone turnover and blood supply in comparison to cortical bone [34,35,36]. Young animals may have an increased absorption rate compared to adults, and females may absorb less BP than males [33]. Once in the bone, BPs are absorbed by osteoclasts via endocytosis [23,25]. As bone resorption continues, BPs reactivate, resulting in prolonged osteoclastic inhibition over time [20]. For this reason, BPs’ half-life is difficult to define and may be up to 10 years depending on species, age, and the specific BP [4,28].

Clodronate and etidronate (sBPs) can undergo intracellular metabolization [37]; on the other hand, there is no evidence of metabolization of nBPs [36]. BPs not taken up by the bone are largely excreted unaltered by the kidneys [28,36]. In human and rat studies, BPs half-life ranges between 1 to 2 h, a quick bloodstream elimination that depends on kidney excretion (renal clearance) and bone uptake (nonrenal clearance); this ratio varies among BPs. In humans, the clodronate renal/nonrenal clearance ratio is between 1.8 to 3 in comparison to pamidronate, which is 0.18 [28]. Additionally, the same class of BPs may differ in some PK parameters. For example, clodronate and tiludronate are sBPs with similar potency [22], yet the half-life of clodronate is reported to be between 2 and 3 h [38,39], and tiludronate’s half-life is much longer (51 h) [40]. Therefore, it is not accurate to extrapolate drug properties, even if BPs belong to the same class [41]. Because of this, studies must evaluate each BP and its individual effects on bone modeling and remodeling.

## 5. Therapeutic Effects of Bisphosphonates

Human conditions treated with BPs include postmenopausal osteoporosis, Paget’s disease, osteogenesis imperfecta, and bone cancer/metastasis [42,43]. Bisphosphates are used to decrease bone resorption and increase bone mineral density (BMD) by decreasing bone catabolism.

In addition to their main antiresorptive properties, BPs are believed to have anti-inflammatory and analgesic effects [4,44,45,46,47,48], which make them an attractive potential treatment for multiple diseases, including osteoarthritis (OA). Meta-analyses and systematic reviews have concluded that BP studies have controversial results, and BP effects may be more related to pain relief than disease modification [48,49,50]. These pain-relieving effects could be beneficial for some individuals, but in human or animal athletes, masking pain could also be dangerous and lead to further deterioration of joint conditions.

BPs anti-inflammatory and pain-relieving effects may be due to a reduction in inflammatory mediators, such as Prostaglandin E_2_ (PGE_2_). PGE_2_ is considered one of the key inflammatory mediators, generating pain in OA [51], and is a critical outcome measure in equine OA studies [52,53]. There is evidence that people treated with neridronate, a nBPs, for osteogenesis imperfecta have reduced serum concentrations of PGE_2_ and CTX-I/creatinine ratio [44]. Equine studies have demonstrated pain relief from BP administration in back OA [54], lower hock osteoarthritis [55], and navicular syndrome [56]. However, it is still unclear how BPs decrease pain in horses [53]. Additional pain-relieving mechanisms may be involved.

Recently, clodronate was identified as a selective and potent inhibitor of vesicular nucleotide transport (VNUT) [57]. Typically, this transporter is responsible for storing ATP in neurons. Research suggests clodronate may inhibit the release of ATP acting as a presynaptic blocker attenuating chronic neuropathic pain [57]. This activity may explain why in an equine study, clodronate did not change serum bone resorption markers but did significantly improve lameness [56]. Further investigation is warranted, as masking pain could lead to adverse outcomes, especially in exercising animals. This is especially important, as bisphosphonates may remain active in the bone for months to years following administration.

## 6. Bisphosphonates’ Side Effects: Humans and Animals

Bisphosphonates have short- and long-term side effects. In humans, the kidneys excrete about 50 to 60% of BPs without major biotransformation [21,22,25], and a rapid intravenous infusion can produce focal glomerulosclerosis [58]. Humans treated with BP can experience short-term adverse effects, including fever, muscle aches, vomiting, and transient hypocalcemia [59]. Long-term exposure in humans can result in serious side effects, including osteonecrosis of the jaw (ONJ) [60] and atypical femur fracture (AFF) [61]. In horses, short-term side effects may include renal toxicity, especially when the animal has a history of renal disease or has been treated with nonsteroidal anti-inflammatory drugs [47]. Further, transient colic-like symptoms have been documented following intravenous infusion [22,38,40]. Other adverse side effects have not been well documented in horses, but a lack of documentation may be related to the scarcity of long-term studies currently available.

Between one and twelve percent of human oncologic patients experience ONJ after three years of IV treatment with BPs [62]. This condition has also been reported in several animal models, including sheep [63,64], mini-pigs [65], and dogs [66]. Recently, clinical reports showed symptoms compatible with ONJ in cats treated with long-term BPs for idiopathic hypercalcemia [67,68]. An additional serious adverse side effect, AFF, accounts for 1.1% of all femur fracture cases in humans [69]. Likewise, a bilateral patellar fracture has been reported in a cat after 8 years of alendronate treatment [70]. Changes in the femoral neck with increased bone brittleness have been found in mice using ibandronate [71]. Even though there is not currently a clear connection between equine stress fractures or catastrophic injuries and BPs, these drugs have shown the potential to produce severe adverse effects in multiple animal models and humans. In horses, clodronate has been isolated from bone samples 18 months following a single administration [72], and tiludronate has been found in low concentrations in plasma (0.05–1.0 ng/mL) and urine samples (0.03–1.5 ng/mL) after three years following administration [73]. Fluctuations in plasma and urine concentrations over time may have been influenced by activity level, health status, growth, and animal-to-animal variation [73]. BPs can be present in the skeleton of horses for long periods of time, potentially masking pain, and are documented to cause adverse bone effects in multiple species. Consequently, further investigation into the relationship between BPs and bone injuries in horses is crucial for equine health.

## 7. Bisphosphonate in Adult Horses

Two BPs were approved for use in horses by the FDA in 2014 for the treatment of navicular syndrome in horses over 4 years old [4]. Clodronate and tiludronate dosage characterizations are detailed in their respective FDA Freedom of Information Summary [38,74]. The lowest effective clodronate dosage found to decrease one grade in navicular-syndrome-associated lameness was 1.8 mg/kg or 900 mg per horse [73]. For tiludronate, 1 mg/kg was found to alleviate symptoms associated with navicular syndrome [38,75]. BPs have resulted in reduced pain and lameness in other skeletal conditions, such as back pain, lower hock OA, and fetlock OA [54,55,76]. Additionally, BPs have been used to treat bone fragility disorder, an osteoclast-mediate osteoporosis [77,78]. Multiple publications have focused on short-term benefits of bisphosphonate use in horses. However, long-term studies investigating the potential long-term adverse effects are lacking.

During the 2019 American Association of Equine Practitioners convention, a roundtable discussion covered the extra-label use of BPs by equine practitioners. Participants indicated BPs were being used for various conditions with radiographic or nuclear scintigraphic abnormalities of the sacroiliac area, pelvis, or limb [79]. Participants described frequent BP administration (e.g., three full doses in a month) despite the manufacturer’s recommendation of a six-month separation between the doses [38]. Researchers have raised concern about the extra-label use of BPs [5], especially in younger horses, where bone turnover is significantly higher in individuals under 24 months of age [80].

## 8. The Use of Bisphosphonates in Young/Exercising Animals

Racehorses often start training and racing at 2 years of age. There is evidence that improper training and management is more of a factor in skeletal injury than age [81], as high-performance exercise may result in progressive microdamage accumulation, potentially leading to stress fractures (SF) [82]. Stress fractures have been associated with a high remodeling rate, leading to bone weakness and accumulation of microdamage over time [83]. It is believed BPs may be useful in preventing athletic SF due to their antiresorptive properties [84]. However, there is no conclusive evidence indicating SF healing by BPs [85], and their use in this condition is not recommended [86]. In truth, bone modeling and remodeling are complex processes, especially when growth and exercise intersect. SF have been associated with normal remodeling and high strains, or normal strains with decreased remodeling [87]. Even though it is not clear what pathophysiological mechanism prevails in racehorses, any interruption in normal osteoclast resorption could be harmful and lead to damage accumulation over time.

In horses, common skeletal conditions, such as dorsal metacarpal disease, commonly known as bucked shins, and sesamoiditis, have been treated with BPs for their perceived skeletal and analgesic effects [4]. Dorsal metacarpal disease results from microfractures on the metacarpal cortical area, and sesamoiditis is the result of disease or osteolysis of the vascular channels of the proximal sesamoid bones [5]. It is believed that BPs may prevent pain and radiographic evidence of these pathologies [4,5]. However, the resolution of radiographic evidence of disease may be accompanied by detrimental effects in juvenile horses, where increasing bone density may not equate to increased bone strength. Further, impairing osteoclast function may harm normal bone remodeling and healing necessary for juvenile horses under high-performance exercise [5].

Bone turnover can be affected by exercise, and BPs can influence physiological adaptation to exercise. Bone resorption increases in response to acute exercise [88]. In long-term exercise, there is a BMD increase, indicating that prolonged exercise can be an osteogenic stimulus [88]. The increased mechanical load under exercising conditions induces osteoclast activation that can result in increasing serum markers of bone remodeling, such as CTX-I [88]. However, serum bone remodeling markers are not strong predictors of bone formation and/or resorption in human subjects [88]. For example, calves subjected to sprints 1 to 5 times per week have increased fracture force and dorsal width of their fused metacarpus compared to a non-exercise group, but no differences in CTX-I were detected between the groups [89]. On the other hand, procollagen type II C-propeptide (CPII) and CTX-I increase as a response to exercise and bone turnover in foals [90]. In humans, BPs may reduce serum bone markers over time [91,92,93]. In horses, conflicting reports exist regarding the effect of BPs on CTX-I [40,56,76,94]. In conclusion, BPs may alter the normal skeletal adaptation to exercise, and assessment of the antiresorptive effects of BPs through serum bone markers is likely insufficient if performed alone. Future studies should consider new, comprehensive approaches to evaluate BP effects, including measuring bone mineral density, fracture healing, and biomechanical testing, while simultaneously determining BP concentration within the bone. In addition, advanced imaging, such as micro-computed tomography (µCT) and positron emission tomography (PET) CT, may be warranted.

The use of BPs may have a greater impact on young horses due to their active growth, where osteoclasts play a significant role in the endochondral ossification process [13,19]. Osteoclasts are abundantly present in growing epiphyseal plates up to 2 years old [80]. The extra-label use of BPs in young animals could impair physiological bone development in this population [80,95]. This has been demonstrated in a rabbit model, where BP administration caused a 3% decrease in the length of the tibia [96]. Hence, BPs use in young animals could pose a significant risk to skeletal growth and/or adaptation to exercise, resulting in microdamage accumulation in juvenile horses without degenerative bone disorders.

## 9. Bisphosphonates and Future Studies

Multiple animal models have already been used to investigate BP, including mice, rabbits, mini-pigs, dogs, and sheep [63,64,65,66,71,96]. The authors recognize the ethical concerns around using animals for research purposes. However, some animal models may be particularly useful depending on the research goals and prior studies available. In particular, the sheep model has proven to be a reliable orthopedic model for human BP use. Sheep have a similar body weight and skeletal size to humans. Procedures such as bone biopsies and blood sampling are simple; they are easy to handle, and large numbers of animals are usually available [97,98,99,100]. Furthermore, sheep can be trained to undergo forced exercise [101], making sheep a suitable animal model for investigating potential BP-associated bone changes under different exercise regimens. Although animal models have been used to investigate long- and short-term BP effects with a focus on human health, few studies are available to guide equine use, especially in juvenile and exercising populations. Future studies may include experimental large animal models of BP use, which incorporate exercise to mimic athletic training. Specifically, terminal ovine models may allow for mechanical testing, advanced imaging, and analysis of long-term BP retention in bone and other organs. These studies, coupled with focused equine experimental trials, prospective and retrospective studies, would provide a more comprehensive explanation of the benefits and risks of BP use in horses.

To date, a single, large, retrospective study has evaluated the efficacy and safety of tiludronate in 1804 horses; 343 horses were followed for over 1 year [102]. The study revealed a low incidence of short-term adverse effects (1.3%), with colic-like symptoms being the most frequent. Less than 20% of horses were treated for navicular syndrome, confirming the extra-label use of BPs. Between one and nine doses of tiludronate were administered to horses included in the study [102]. Treated horses ranged in age from 2 years old to 26 years old. Future retrospective studies would ideally report diagnosis, age at administration, number and frequency of doses, long-term follow-up, concurrent treatments, and evidence of disease progression.

Future prospective studies will ideally look beyond serum biomarkers and report multiple clinical and experimental parameters. These could include physical and lameness examinations coupled with bone biopsies, synovial fluid analysis, advanced imaging, and biomechanical testing. Veterinarians, owners, and researchers alike would benefit from a better understanding of the half-life of BPs within the skeleton and the physiologic factors, such as age and exercise, which may change the half-life of BPs. Tiludronate has been previously measured in tuber coxae biopsies, a relatively non-invasive location for bone biopsy [103]. Tiludronate can be detected with ultra-high-performance liquid chromatography–high-resolution mass spectrometry for up to three years in plasma and urine samples, and clodronate was detected in bone 18 months following administration in a single horse in a single study [72,73]. However, the long-term presence of BPs in bone in a large clinical population is currently unreported [72]. Little information is currently available to guide the frequency of dosing to ensure clinical efficacy and safety.

The pain-relieving effects of BPs are still being investigated. Although pain relief may be a clinical benefit, it could also result in further injury, especially in high-performance athletes. BPs have been detected in the synovial fluid after systemic administration [39]. Further investigation is necessary to understand the potential anti-inflammatory effects of BPs systemically and within the joint environment. This can be accomplished through in vitro studies, in vivo animal models of pain and inflammation, and clinical studies [104].

## 10. Conclusions

Bisphosphonates are well known for their antiresorptive properties, impairing osteoclast functionality. In 2014, two bisphosphonates (clodronate and tiludronate) were approved by the FDA to treat navicular syndrome in horses over four years of age. Several in vitro animal models and human studies indicate that bisphosphonates may have anti-inflammatory and pain-relieving effects, which has led to extra-label use of these drugs for other conditions and in juvenile horses. Although there may be therapeutic effects, there are concerns regarding impairment of normal physiological functions (growth, bone repair, and bone remodeling), especially in juvenile and exercising animals. Additional research must focus on identifying the short-term and long-term effects of bisphosphonates in young and exercising animals to ensure the efficacious and judicious use of this powerful, long-lasting group of drugs.

## Figures and Tables

**Figure 1 animals-12-01722-f001:**
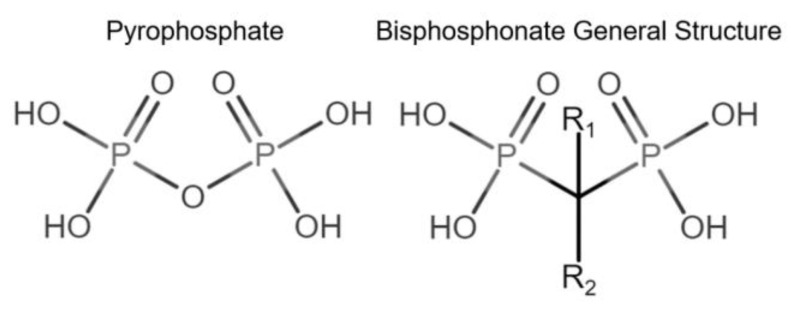
Comparison of pyrophosphate and basic bisphosphonate structures. Bisphosphonates differ from pyrophosphates primarily by the change of oxygen from their central atom to carbon, providing resistance to biological degradation.

**Figure 2 animals-12-01722-f002:**
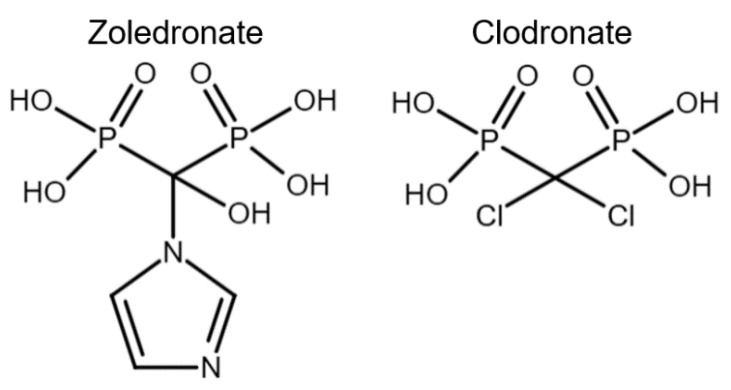
Chemical structure comparison between a third-generation nitrogen-containing BP (zoledronate) and a first-generation non-nitrogen-containing BP (clodronate).

**Table 1 animals-12-01722-t001:** Summary of osteoclastogenesis molecules, origin, and functions. Molecules essential for osteoclastogenesis include: receptor activator of nuclear factor (NF)-ĸB ligand (RANKL), osteoprotegerin (OSP), macrophage-colony stimulating factor (M-CSF).

Molecules	Origin	Function
RANKL	Bone-marrow-derived stem cells, osteoblasts, osteocytes	Primary differentiation factor controlling gene expression binding to RANK [11,12]
OPG	Osteoblast and osteocytes	Decoy receptor for RANKL competing with RANK. Blocks RANKL–RANK interaction [11]
M-CSF	Bone-marrow-derived stem cells, osteoblasts, osteocytes	Activates pathways stimulating proliferation and survival by binding to macrophage colony-stimulating factor 1 receptor (CSF-1 R/c-Fms) [11,12]

## Data Availability

No new data were created or analyzed in this study. Data sharing does not apply to this article.

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
