# Peer review of "Is the Use of Bisphosphonates Putting Horses at Risk? An Osteoclast Perspective"

_animals, 2022, doi:10.3390/ani12131722_

Round 1
Reviewer 1 Report
Comments to the Authors of manuscript number: animals-1779883 entitled “Is the Use of Bisphosphonates Putting Horses at Risk? An Osteoclast Perspective”.
In general, the study design is very interesting, results are presented well. The manuscript is written in quate different style then other presented traditionally due to mathematical aspect. Consistently written and easy to follow.
1. L 28 – the age of the skeletally-immature animals have to given
2. L 30 – juvenile horses are skeletally immature
3. The introduction shortly provided the main goal of the paper
4. L 63- is OPG/RANK/RANKL system key in the muscle development? It is a key in bone modeling and remodeling
5. L 82 It runs on two different surfaces, leading to the mature of bone, and allow to reach the proper geometry
6. L 84 – it runs on the same surface
7. The part 4 described bisphosphonates very well
8. The part 5 also presents therapeutic properties of bisphosphonates very well
9. The part 6. The journal of Animals is rather the journal relating to animals. I recommend to short the part of the text about the side effects in humans, and to focus on animals
10. this comment is linked with the previous point. These parts describing the side effects, the use of bisphosphonates in adult and young horses, connect into one part, omitting people
11. the review is very interesting and worth for publication
Author Response
Reviewer 1
In general, the study design is very interesting, results are presented well. The manuscript is written in quate different style then other presented traditionally due to mathematical aspect. Consistently written and easy to follow.
Response: The authors appreciate the time invested in this review, the helpful comments and corrections by the reviewer. We believe, we have improved the quality and clarity of the review.
- L 28 – the age of the skeletally-immature animals have to given
Response: We appreciate this comment and we have included the age. Now the sentence reads: “Currently, there is a critical gap in scientific knowledge surrounding the biological impacts of BP use in exercising animals under two years old.” (Lines 27-28)
- L 30 – juvenile horses are skeletally immature
Response: This sentence has been altered to avoid confusion. Now the sentence reads: “Therefore, future research should investigate the effects of these drugs in skeletally immature horses.” (Lines 29-30)
- The introduction shortly provided the main goal of the paper
Response: The authors appreciate your positive comment about our introduction’s main goal.
- L 63- is OPG/RANK/RANKL system key in the muscle development? It is a key in bone modeling and remodeling
Response: We apologize for this misunderstanding. We have addressed this issue and now the sentence reads: “Osteoclastogenesis and subsequent osteoclast-mediated resorption are necessary for skeletal development, bone remodeling, and bone repair.” (Line 62-63).
- L 82 It runs on two different surfaces, leading to the mature of bone, and allow to reach the proper geometry
Response: We thank you for this comment. We have complemented the review with your points and the sentence now reads: “Bone modeling is the process of mineral uptake and removal in growing organisms, leading to bone maturation. Meanwhile, bone remodeling is a process that takes place throughout the life of organisms as the bone adapts to new mechanical loads and re-pairs microdamage, allowing the bone to reach a proper geometry [19].” (Lines 83-86)
- L 84 – it runs on the same surface
Response: We apologize but we would like more clarity from the reviewer about this comment.
- The part 4 described bisphosphonates very well
Response: We appreciate the positive comment from the reviewer.
- The part 5 also presents therapeutic properties of bisphosphonates very well
Response: Thank you for agreeing with the information proportioned in the review.
- The part 6. The journal of Animals is rather the journal relating to animals. I recommend to short the part of the text about the side effects in humans, and to focus on animals
Response: Even though the authors recognize that is an animal focused journal, we feel there is value to some information regarding human side effects especially because limited studies are available in animals. As mammals, human subjects have been treated for several decades with bisphosphonates resulting in more data in human species. Some of the negative side effects in humans have already been reported in some animal species (despite limited studies) including kidney effects shared in humans/horses (line 188, 192-194) and osteonecrosis of the jaw (line 198-200). We believe negative side effects may be currently under-reported in animal species. For these reasons, we would like to keep this information within the manuscript.
- this comment is linked with the previous point. These parts describing the side effects, the use of bisphosphonates in adult and young horses, connect into one part, omitting people
Response: We acknowledge that talking about humans’ side effects may not be the main target of the manuscript, yet we find worth to highlight those effects and relate them to animals. Recognizing such side effects, we can be careful and monitoring for those effects in animals. Moreover, we value having adult and young horses in separated sections. One of the main purposes of this review is highlight the lack of research in young/exercising horses. Combining both sections – adult and juvenile horses – we may loss the significance that these drugs may produce in juvenile populations of horses.
- the review is very interesting and worth for publication
Response: We appreciate your comments and have done our best to address them all. Thank you for your constructive review and recommendation for this review to be published.

Reviewer 2 Report
Congratulations to the authors for crafting a comprehensive, well structured overview of pertinent information related to bisphosphonate therapy, osteoclast activity and their potential for adverse effects in young racehorses. This is an important, topical subject and an update will interest Animals’ readership. It was a pleasure to read because of its clarity and logical progression. Even though it is a complex subject, they have covered all the important issues.
Minor suggestions for improvement
Abstract
L1. In respect to osteoporosis add “humans”
L22-23. Consider rewording “extra-label bisphosphonate use could result in skeletal microdamage”. The microdamage includes the cracks that arise in the bone structure. It may “accumulate” if there is no osteoclastic repair. Bisphosphonate therapy does not cause microdamage directly.
L44. Please insert commercial names here.
L 205. Why only catastrophic fractures? It is possible that they could in some way interfere with all stress fractures.
L 209. This section could be expanded to add that peaks in urine concentrations also occurred during the three years suggesting that the release of BPs is infleunced by a variety of factors.
L 242. Current thinking is that microdamage arises in bone first in racehorses and this will result in local osteoclast differentiation and resorption in an attempt to repair the tissues. Whether the microdamage alone, or microdamage combined with resorption, or accumulation of microdamage because of lack of resorption in training and racing leads to bone weakening has not yet been clearly establised. “Osteoclast resorption” may also be more specific.
L 245. Spelling mistake: “are complex processes” (two processes)
L 264-265: Mention that this study is referring to humans.
L291-300. There is an ethical conundrum in this paragraph it would seem. Is it acceptable to euthanatize sheep as a model for the horse? Is a horse superior to a sheep? Even though we, and the public, may view horses more favorably, unfortunately, terminal studies may be required in some groups of horses to accurately address many of these unknowns. I would suggest that you word this section differently to remove the sheep as a model for the horse . The argument does not add much and detracts from the nice work previously.
Author Response
Reviewer 2
Congratulations to the authors for crafting a comprehensive, well structured overview of pertinent information related to bisphosphonate therapy, osteoclast activity and their potential for adverse effects in young racehorses. This is an important, topical subject and an update will interest Animals’ readership. It was a pleasure to read because of its clarity and logical progression. Even though it is a complex subject, they have covered all the important issues.
Response: We appreciate your comments and have done our best to address them below. Thank you for your constructive review.
Minor suggestions for improvement
Abstract
L1. In respect to osteoporosis add “humans”
Response: We appreciate this addition. The sentence now reads: “These drugs are used for skeletal conditions, such as osteoporosis in humans and are available for veterinary medical use.” (Lines 11-12)
“
L22-23. Consider rewording “extra-label bisphosphonate use could result in skeletal microdamage”. The microdamage includes the cracks that arise in the bone structure. It may “accumulate” if there is no osteoclastic repair. Bisphosphonate therapy does not cause microdamage directly.
Response: Thank you for your comment. We recognize that the potential damage is an indirect effect of bisphosphonates so now the sentence reads: “However, extra-label bisphosphonate use may impair osteoclast function which could result in skeletal microdamage and impaired healing without commonly associated pain, affecting bone remodeling, fracture healing, and growth.” (lines 22-25)
L44. Please insert commercial names here.
Response: We have included the commercial names. Now, the sentence reads: “In 2014, two bisphosphonates (Osphos® and Tildren®) were approved by the United States Food and Drug Administration (FDA) to treat navicular syndrome in horses over four years of age [4].” (Lines 44-46)
L 205. Why only catastrophic fractures? It is possible that they could in some way interfere with all stress fractures.
Response: We appreciate this comment. We agree that bisphosphonates will not only produce catastrophic injuries. We have corrected the sentence and it reads now: “Even though there is not currently a clear connection between equine stress fractures or catastrophic injuries and BPs, these drugs have shown the potential to produce severe adverse effects in multiple animal models and humans.” (Lines 206-208)
L 209. This section could be expanded to add that peaks in urine concentrations also occurred during the three years suggesting that the release of BPs is infleunced by a variety of factors.
Response: We have included the information regarding plasma and urine concentrations of the reference study, also including a comment about possible influential factors in these bisphosphonates’ fluctuations over time. The sentences now read: “…, and tiludronate has been found in low concentrations in plasma (0.05 – 1.0 ng/ml) and urine samples (0.03 – 1.5 ng/ml) after three years following administration [74]. Fluctuations in plasma and urine concentrations over time may have been influenced by activity level, health status, growth, and animal to animal variation [74].” (Line 209-213)
L 242. Current thinking is that microdamage arises in bone first in racehorses and this will result in local osteoclast differentiation and resorption in an attempt to repair the tissues. Whether the microdamage alone, or microdamage combined with resorption, or accumulation of microdamage because of lack of resorption in training and racing leads to bone weakening has not yet been clearly establised. “Osteoclast resorption” may also be more specific.
Response: We appreciate and agree with this comment. Although line 242 talks about the previous rationale to use bisphosphonates in stress fractures, in the following sentences we state that the use of BP may be actually counterproductive. We have included new language in the paragraph. This now reads: “SF have been associated with normal remodeling and high strains, or normal strains with decreased remodeling [89]. Even though it is not clear what pathophysiological mechanism prevails in racehorses, any interruption in normal osteoclast resorption could be harmful and lead to damage accumulation over time.” (Line 248-252).
L 245. Spelling mistake: “are complex processes” (two processes)
Response: Thank you for finding this spelling mistake. It has been corrected (Line 248)
L 264-265: Mention that this study is referring to humans.
Response: We have included the reference to humans. This reads: “However, serum bone remodeling markers are not strong predictors of bone formation and/or resorption in human subjects [90].” (Line 268-269)
L291-300. There is an ethical conundrum in this paragraph it would seem. Is it acceptable to euthanatize sheep as a model for the horse? Is a horse superior to a sheep? Even though we, and the public, may view horses more favorably, unfortunately, terminal studies may be required in some groups of horses to accurately address many of these unknowns. I would suggest that you word this section differently to remove the sheep as a model for the horse . The argument does not add much and detracts from the nice work previously.
Response: We appreciate your comment and recognize the constant ethical dilemma that implies using experimental animal models for research purposes. By no means do we believe that sheep are an inferior species to horses, or to other research animals. Nevertheless, their use/purpose (meat, wool, and/or research), often ends in slaughter. This is contrary to horses where slaughter is forbidden in the US, and their main purpose is for sports, recreation, and as a companion animal. Moreover, several studies have used the ovine model to characterize the effects of bisphosphonates on humans. This could lead to potential translational results that can be used not only for equine health but also for human health. Another reason to use an animal model for horses is to generate preliminary information that can be used as a basis for further specific research on the horse, avoiding unnecessary sacrifice of horses. Complementary advantages of using sheep are their more homogenous genetical background, cost-effective maintenance, and ease of sampling among other factors. For all these reasons we believe that there are objective advantages of proposing this particular animal model to study bisphosphonates effects in horses in comparison to other available animal models.
We hope you can understand our perspective and we appreciate the reviewer’s concern. In an effort to recognize the ethical concerns around research animals we have included new language: “Multiple animal models have already been used to investigate BP including mice, rabbits, mini-pigs, dogs, and sheep [64–67,72,98]. The authors recognize the ethical concerns around using animals for research purposes. However, some animal models may be particularly useful depending on the research goals and prior studies available. In particular, the sheep model has proven to be a reliable orthopedic model for human BP use. Sheep have a similar body weight and skeletal size to humans, procedures such as bone biopsies and blood sampling are simple, they are easy to handle, and large numbers of animals are usually available [99–102]. Furthermore, sheep can be trained to undergo forced exercise [103], making sheep a suitable animal model for investigating potential BP-associated bone changes under different exercise regimens. (Lines 294-303)
We have also expanded language to indicate that further, targeted, equine studies are necessary “These studies, coupled with focused equine experimental trials, prospective and retrospective studies would provide a more comprehensive explanation of the benefits and risks of BP use in the horse.” (Line 309-311)
